# Identification of the Constitutive Model Parameters by Inverse Optimization Method and Characterization of Hot Deformation Behavior for Ultra-Supercritical Rotor Steel

**DOI:** 10.3390/ma14081958

**Published:** 2021-04-14

**Authors:** Xuewen Chen, Yuqing Du, Kexue Du, Tingting Lian, Bingqi Liu, Zhipeng Li, Xudong Zhou

**Affiliations:** School of Materials Science and Engineering, Henan University of Science and Technology, 263 Kaiyuan Avenue, Luoyang 471023, China; DuyqStephanie@163.com (Y.D.); 18437955329@163.com (K.D.); lian_tingting@126.com (T.L.); lbq8565@163.com (B.L.); mrlzp597@163.com (Z.L.); syuuzhou@163.com (X.Z.)

**Keywords:** X12 ferritic heat-resistant steel, Hansel–Spittel constitutive model, reverse optimization method, characteristic area distribution map, the optimal processing parameters

## Abstract

X12 (X12CrMoWVNbN10-1-1) ferritic heat resistant steel is an important material for the production of new-generation ultra-supercritical generator rotors. Hot compression tests of X12 ferritic heat-resistant steel were performed via a Gleeble-1500D testing machine under temperatures of 1050–1250 °C and strain rates of 0.05–5 s^−1^. In order to provide material model data for finite element simulations and accurately predict the hot deformation behavior, a reverse optimization method was proposed to construct elevated temperature constitutive models of X12 ferritic heat-resistant steel in this paper, according to the Hansel–Spittel constitutive model. To verify the accuracy of the model, the predicted and experimental values of the constitutive model were compared. The results indicated that the model had a high prediction accuracy. Meanwhile, the correlation coefficient between the experimental value and the predicted value of constitutive model was 0.97833. For further verification of the accuracy of the model, it was implemented in finite element FORGE^@^ software to simulate the compression tests of different samples under different conditions. Comparing actual displacement–load curves with displacement–load curves acquired through finite element simulations, the results indicated that displacement–load curves predicted by the model were very consistent with actual displacement–load curves, which verified the accuracy of the model. Moreover, to research the optimal processing parameters of the material, hot processing maps were drawn according to the dynamic material model. In terms of microstructure evolution, a characteristic area distribution map of the hot processing map was established. Therefore, the optimal hot forming parameters regions were in the range of 1150–1200 °C/0.05–0.62 s^−1^ for X12 ferritic heat-resistant steel.

## 1. Introduction

In order to reduce environmental pollution and greenhouse gas (CO_2_) emissions, ultra-supercritical power generation technology has become the main power generation mode of thermal power plants [1,2]. High and medium-pressure rotors are the crucial component of ultra-supercritical unit. As a large forging workpiece, its hot processing procedure is complicated and the manufacturing time is very long. It is difficult to guarantee the quality of the product because defects in the structure and cracks often appear in the manufacturing process. Thus, it is of great importance to research the deformation characteristics of steel (X12 ferritic heat-resistant steel) for manufacturing ultra-supercritical rotors in elevated temperatures. X12(X12CrMoWVNbN10-1-1) steel is based on 9Cr1Mo steel (X10CrMoBNb91) with an added alloying element, W, in the European COST51 plan, which is a typical ferritic heat-resistant steel [3].

Since the development and application of constitutive models are particularly important in terms of finite element numerical simulations, the production process of workpieces can be simulated through finite element software. This method can optimize the process parameters, which is of great significance to shorten the production cycle and reduce costs. In recent years, many scholars have researched the hot deformation behavior of materials by establishing constitutive models and hot processing maps, combined with microstructure evolution. The Johnson–Cook model was proposed by Johnson and Cook [4], which has a clear physical meaning and is mainly used to build flow stress models under conditions of elevated temperature, high strain rate, and large strains. However, it is not suitable for research of hot deformation behavior with low strain rates, as well as small strains. Laasroaui and Jonas [5,6] used single-pass compression tests on low-carbon steel and then established a two-stage model to study the hot deformation behavior of low-carbon steel. Although the two-stage model of dynamic recovery, work hardening, and dynamic recrystallization can express the effects of strain on stress, there are many parameters in the model, which are relatively complex. Sellars [7,8,9,10] established a hyperbolic sinusoidal (Arrhenius flow stress model) material constitutive model and considered the relationship between strain rate and steady-state stress. As one of the common constitutive models, the Arrhenius flow stress model is divided into hyperbolic sine, exponential, and power exponential equations, which are applied to materials under various stresses. However, strain softening is not considered in this model. The stress value before steady state cannot be determined. When using finite element software, the selection of a material model and the setting of relevant parameters play a decisive role in the accuracy of test results. Compared with other models, the Hansel–Spittel constitutive model has a wider range of application. The model contains nine parameters, namely A, $m_{1}$, $m_{2}$, $m_{3}$, $m_{4}$, $m_{5}$, $m_{6}$, $m_{7}$, $m_{8}$ and $m_{9}$, which are the relevant factors of each item of the corresponding formula [11,12]. As an important flow stress model [13,14], it is widely used to delineate the hot and cold forming behaviors of materials in finite element software, such as FORGE^@^ or QFORM.

One major issue is the accurate identification of the parameters of a material constitutive model, which will influence the accuracy for the simulation results for the hot forming process. As for the method of determining the parameters of a material constitutive model, a great deal of existing the literature adopted the traditional calibration method [15,16,17]. For example, when solving the parameters of a constitutive model, such as hyperbolic sine function, a variable is first made a constant value. Then, the parameter values are solved under various deformation conditions. Finally, their average is taken. The model is as follows:(1)$$Z\;=\;\dot{\varepsilon }\exp[Q/(RT)]\;=\;A{\left(\sinh(\alpha \sigma )\right)}^{n}$$

T is first made constant. Then, the logarithm of both sides of Formula (1) is taken. Formula (2) is then obtained as follows:(2)$$\ln\dot{\varepsilon }\;+\;Q/RT\;=\;n\ln(\sinh(\alpha \sigma ))\;+\;\ln A$$

Therefore, Formula (3) can be acquired as follows:(3)$$n\;=\;{\left[\frac{\partial \ln\dot{\varepsilon }}{\partial (\ln(\sinh(\alpha \sigma )))}\right]}_{T}$$

The final value is the average of n values under various conditions. If each parameter is calculated according to this method, this will greatly increase the error of the constitutive model, which affects the precision of numerical simulation results. Recently, due to the development of the optimization algorithm, it is possible to solve the parameters of the complex material constitutive model. The reverse analysis method is often utilized to identify material parameters. The essence of parameter reverse analysis is to select an appropriate optimization algorithm to continuously adjust the parameters and make the optimization target approach the test target, based on the physical quantity, which is easy to measure. At present, there is little research on the application of reverse optimization technology for parameter identification of constitutive models. Only some researchers have applied this method in the field of solving complex damage model parameters. Specifically, Abbasi [18] used the response surface method to identify GTN(Gurson-Tvergaard-Needleman) damage model parameters of interstitial-free atomic steel. Then, the optimized damage model parameters were input into ABAQUS to predict the damage behavior of materials. Lauro [19] put forward a damage–mechanics model of void materials, considering the influence of strain rates. The parameters of the damage model are defined by a reverse engineering algorithm. Based on the above analysis, the method to identify elevated temperature constitutive model parameters of X12 ferritic heat-resistant steel using a reverse optimization algorithm is proposed in this paper. Combined with the finite element method (FEM), the deformation behavior of X12 ferritic heat-resistant steel can be accurately predicted at elevated temperatures.

For the sake of further researching the optimal forming process of ultra-supercritical rotors, a hot processing map based on a dynamic material model is used in this paper. Prasad and Gegel established a dynamic material model (DMM) of Ti-6242 alloy on the basis of the irreversible thermodynamics theory, continuum mechanics of large plastic deformation, and physical system simulations to depict the hot deformation behavior of the material [20]. With the deepening of research, scholars began to construct material constitutive equations and hot processing maps to describe the microstructure evolution mechanism and hot deformation behaviors. Specifically, Wang et al. established the Avrami constitutive equation of 30Si2MnCrMoVE low-alloying ultra-high-strength steel to depict its deformation behavior, based on experimental data. Moreover, a hot processing map of the material was constructed [21]. Zhao et al. constructed an Arrhenius constitutive equation considering strain compensation of 1Cr12Ni2Mo2WVNb martensitic stainless steel. In addition, the forgeability of 1Cr12Ni2Mo2WVNb steel was researched. The best process conditions for 1Cr12Ni2Mo2WVNb steel were temperatures of 1070–1100 °C and strain rates of 0.8–1 s^−1^ [22].

In this paper, a new method for determining constitutive model parameters was proposed, based on reverse optimization technology. An elevated temperature flow stress model of X12 ferritic heat-resistant steel was constructed based on experimental data acquired on a Gleeble-1500D testing machine at strain rates of 0.05–5 s^−1^ and deformation temperatures of 1050–1250 °C. Combined finite element technology, the constitutive model was embedded into FORGE^@^ software to predict the elevated temperature deformation behavior of X12 ferritic heat-resistant steel. The results indicate that the displacement–load curve predicted by the model is consistent with the actual displacement–load curve, which verified the accuracy of the model. In addition, in order to further research the optimal forming range of ultra-supercritical rotors, combined with the dynamic material model, the characteristic area distribution of a hot processing map of the material was established on the basis of microstructure analysis. Meanwhile, optimal process parameters of X12 ferritic heat-resistant steel were acquired as follows: strain rate ranges of 0.05–0.62 s^−1^; temperature ranges of 1150–1200 °C, which provide data support for the hot forging process of ultra-supercritical rotors.

## 2. Experimental Materials and Methods

The experimental material was X12 ferritic heat-resistant steel, of which the detailed chemical composition (mass fraction, %) is listed in Table 1. The ingot was processed into a cylinder of 8 mm × 12 mm. Compression tests were performed on a Gleeble-1500D thermal simulation testing machine (Dynamic Systems Inc, New York, NY, USA), as presented in Figure 1. Due to the friction between dies and the specimen in the test machine, the specimen easily produced an obvious drum shape when deformed. To make the pressure distribution uniform, as well as reduce the influence of friction on the test results, both ends of the specimen were coated with graphite for lubrication before compression. The tests were carried out at strain rates of 0.05 s^−1^, 0.1 s^−1^, 0.5 s^−1^, 1 s^−1^ and 5 s^−1^, and the deformation temperatures were 1050 °C, 1100 °C, 1150 °C, 1200 °C and 1250 °C with height compression of 50%. The experiments were divided into 25 groups. After deformation, the sample was immediately quenched and cut using a wire cutting machine. Soon afterwards, the cut surface was grinded with sandpaper and finally the cut surface was polished. The polished surface was corroded with 10% sulfuric acid and 1.25 g potassium permanganate in a 100-mL water bath at 75 °C for 15 s. After corrosion, the surfaces were cleaned with dilute oxalic acid and alcohol. Finally, the microstructure was observed under a metallography microscope (OLYMPUS PMG3, OlympusCorporation, Tokyo, Japan) and scanning electron microscope.

## 3. Experimental Result Analysis of X12 Ferritic Heat-Resistant Steel 

### 3.1. Influence of Process Parameters on Flow Stress

Figure 2 shows the flow stress curves of X12 ferritic heat-resistant steel under various deformation conditions, indicating that the changing trend of flow stress curve is basically consistent. Additionally, flow stress increases greatly as the strain continually increases during the initial stage. This is because as the plastic deformation and dislocation density increase, and the dislocation intercross increases, which produces a dislocation pile-up group, dislocation jog, and dislocation tangle. This will increase the dislocation motion resistance, thereby resulting in an increase in deformation resistance and significant work hardening [24,25,26]. When strain reaches the critical value, dynamic recrystallization begins to occur, of which the softening effect increases gradually as the strain increases. Therefore, the amplitude of stress decreases with strain increases gradually. When the work-hardening rate of X12 ferritic heat-resistant steel is equal to the softening rate of dynamic recrystallization, the flow stress reaches the peak. As dynamic recrystallization is the dominant mechanism, the flow stress gradually decreases until the softening effect and work hardening are in a dynamic balance in which the flow stress starts to stabilize, showing typical DRX(Dynamic recrystallization) characteristics [27,28]. When the temperature is constant, flow stress gradually increases as strain rate rises from 0.05 s^−1^ to 5 s^−1^. For instance, at a temperature of 1100 °C, peak stress increases by 69.65 MPa when strain rate incraeses from 0.05 s^−1^ to 5 s^−1^. The reason is that the higher the strain rate is, the stronger the work hardening effect of the alloy will be in the plastic deformation process. Meanwhile, the dislocation structure can be formed rapidly. Therefore, the dislocation density increases, resulting in insufficient time to move to a favorable position for the intergranular slip in the alloy. Ultimately, the dynamic recrystallization softening effect decreases and flow stress increases. As the strain rate is constant, the flow stress will decrease with the increase in deformation temperature. For instance, at a strain rate of 0.1 s^−1^, deformation temperature increases from 1050 °C to 1250 °C and peak stress will decrease by 89.41 MPa. This is due to the fact that as the increase in deformation temperature occurs, thermal activation of the alloy is enhanced, which increases the dynamic softening rate and intensifies the movement of atoms in the alloy. As a result, the critical shear stress and the binding force between atoms are both reduced, which increases movable slip systems, resulting in reduced flow stress [29,30].

Figure 3 shows the metallographic structure of X12 ferritic heat-resistant steel in the initial state and after hot deformation. It can be observed from Figure 3a that the heat-deformed material has coarse grains and a structure of equiaxed grain before deformation. The average grain size is 60.27 μm. When the strain reaches 0.7, the grain diameter is obviously smaller than the diameter of the initial grain. Figure 3b shows the metallographic structure of X12 ferritic heat-resistant steel at 1100 °C/0.05 s^−1^. It can be observed that the original grains have been replaced by equiaxed grains of different sizes, indicating that complete dynamic recrystallization has taken place under these conditions. Under these conditions, the average grain size is 17.65 μm. The metallographic structure of the material at 1100 °C/0.5 s^−1^ is shown in Figure 3c. It can be seen that grains are greatly deformed and elongated along the vertical force direction. Meanwhile, some dynamic recrystallized grains can be observed around the elongated grains. The reason for this is that a high strain rate will restrict the growth of these small grains, which are only distributed around the original grains. It is proved that incomplete dynamic recrystallization occurs under these conditions. Under these conditions, the average grain size is 12.52 μm. Figure 3d shows the metallographic structure of the material under the conditions of 1250 °C and 0.5 s^−1^, which shows the distribution of equiaxed crystal after growth. This is because, with increasing temperature, the thermal activation energy of X12 ferritic heat-resistant steel increases and the dynamic softening influence is enhanced. Therefore, the growth rate of the dynamic recrystallization grain increases and grain size increases. Under these conditions, the average grain size is 27.69 μm.

### 3.2. Hansel–Spittel Constitutive Model of X12 Ferritic Heat-Resistant Steel

#### 3.2.1. Identification of Hansel–Spittel Constitutive Model Parameters Using Traditional Calibration Method

The Hansel–Spittel model can delineate the thermal deformation behavior of X12 ferritic heat-resistant steel by considering three deformation parameters (strain rate, strain, and temperature) accurately, and is described by the following formula (4):(4)$$\sigma \;=\;Ae^{m_{1}T}\varepsilon^{m_{2}}{\dot{\varepsilon }}^{m_{3}}e^{\frac{m_{4}}{\varepsilon }}{\left(1\;+\;\varepsilon \right)}^{m_{5}T}e^{m_{7}\varepsilon }{\dot{\varepsilon }}^{m_{8}T}T^{m_{9}}$$
where σ represents the equivalent stress, ε represents the equivalent strain and e represents the natural constant. A, $m_{1}$, $m_{2}$, $m_{3}$, $m_{4}$, $m_{5}$, $m_{7}$, $m_{8}$ and $m_{9}$ are the material coefficients.

In this paper, the constitutive model of Hansel–Spittel was constructed using the univariate method. Taking the logarithm of both sides of Equation (4), Formula (5) is acquired as follows:(5)$$\ln\sigma \;=\;\ln A\;+\;m_{1}T\;+\;m_{2}\ln\varepsilon \;+\;m_{3}\ln\dot{\varepsilon }\;+\;\frac{m_{4}}{\varepsilon }\;+\;m_{5}T\ln(1\;+\;\varepsilon )\;+\;m_{7}\varepsilon \;+\;m_{8}T\ln\dot{\varepsilon }\;+\;m_{9}\ln T$$

When the strain and temperature are constant, $\ln A\;+\;m_{1}T\;+\;m_{2}\ln\varepsilon \;+\;\frac{m_{4}}{\varepsilon }\;+\;m_{5}T\ln(1\;+\;\varepsilon )\;+\;m_{7}\varepsilon \;+\;m_{9}\ln T$ is set to $K_{1}$. The following formula (6) can be obtained:(6)$$\ln\sigma \;=\;K_{1}\;+\;(m_{3}\;+\;m_{8}T)\ln\dot{\varepsilon }$$

Taking $\ln\dot{\varepsilon }$ and lnσ as the abscissa and ordinate, it can be seen from Formula (6) that $\ln\dot{\varepsilon }$ and lnσ have a linear relationship. As shown in Figure 4, taking 1100 °C as an example, the slope is equal to $m_{3}\;+\;m_{8}T$, which has a linear relationship with T under different strain conditions. The slope is equal to $m_{8}$ and the intercept average is equal to $m_{3}$ in Figure 5. Thus, the obtained average value of $m_{3}$ is equal to 0.29724. The obtained average value of $m_{8}$ is equal to 0.000383.

When the strain rate and strain are constant, $\ln A\;+\;m_{2}\ln\varepsilon \;+\;m_{3}\ln\dot{\varepsilon }\;+\;\frac{m_{4}}{\varepsilon }\;+\;m_{7}\varepsilon$ to $K_{2}$ is set to get the following formula (7):(7)$$\ln\sigma \;=\;K_{2}\;+\;[m_{1}\;+\;m_{5}\ln(1\;+\;\varepsilon )\;+\;m_{8}\ln\dot{\varepsilon }]T\;+\;m_{9}\ln T$$

Taking T and lnσ as the abscissa and ordinate, they are fitted with formula $y\;=\;K_{2}\;+\;ax\;+\;m_{9}\ln x$ under various deformation conditions. Figure 6 represents the relationship between lnσ and T under various strains at 0.1 s^−1^. The average value of $m_{9}$ can be obtained as –1.59444. With $a\;=\;m_{1}\;+\;m_{5}\ln(1\;+\;\varepsilon )\;+\;m_{8}\ln\dot{\varepsilon }$, when the strain rate is constant; it can be fitted with formula $y\;=\;m_{5}\ln(1\;+\;x)\;+\;K_{3}$ (where $K_{3}\;=\;m_{1}\;+\;m_{8}\ln\dot{\varepsilon }$) under various strain conditions. The values are $m_{5}\;=\;0.001173$, $m_{1}\;=\;-0.00311$.

When temperature and strain rate are constant, $\ln A\;+\;m_{1}T\;+\;m_{3}\ln\dot{\varepsilon }\;+\;m_{8}T\ln\dot{\varepsilon }\;+\;m_{9}\ln T$ to $K_{4}$ is set to get the following formula (8):(8)$$\ln\sigma \;=\;m_{2}\ln\varepsilon \;+\;\frac{m_{4}}{\varepsilon }\;+\;m_{5}T\ln(1\;+\;\varepsilon )\;+\;m_{7}\varepsilon \;+\;K_{4}$$

lnσ and ε are fitted with the function of $y\;=\;m_{2}\ln x\;+\;\frac{m_{4}}{x}\;+\;m_{5}T\ln(1\;+\;x)\;+\;m_{7}x\;+\;K_{4}$ under various temperatures, where the value of $m_{5}T$ is constant. For instance, as shown in Figure 7, at 1250 °C, the average values of $m_{2}$, $m_{4}$ and $m_{7}$ are calculated for all temperatures. The results are: $m_{2}\;=\;0.120072$, $m_{4}\;=\;-0.00631$, $m_{7}\;=\;-1.46508$.

Solving parameter A. 

Taking the values of $m_{1}$, $m_{2}$, $m_{3}$, $m_{4}$, $m_{5}$, $m_{7}$, $m_{8}$ and $m_{9}$ obtained above, and substituting them into Formula (1), the value of A could be obtained under various deformation conditions: $A\;=\;4.03\;\times \;10^{8}$.

Therefore, solving using the traditional calibration method, the high-temperature thermal deformation constitutive model of X12 ferritic heat-resistant steel is:(9)$$\sigma \;=\;\begin{array}{c} 4.03\;\times \;10^{8} \end{array}e^{-0.00311T}\varepsilon^{0.120072}{\dot{\varepsilon }}^{-0.29724}e^{\frac{-0.00631}{\varepsilon }}{\left(1\;+\;\varepsilon \right)}^{0.001173T}e^{-1.46508\varepsilon }{\dot{\varepsilon }}^{0.000383T}T^{-1.59444}$$

However, when the traditional calibration method is used to identify model parameters, it is necessary to take the average of the parameter values under all deformation conditions, which greatly increases the error. For example, when solving $m_{3}$, as presented in Figure 8, its value fluctuated greatly with a maximum difference of 0.26. Therefore, to avoid this type of error and increase the accuracy of the model, a reverse optimization method was adopted to identify model parameters in this paper.

#### 3.2.2. Identification of Hansel–Spittel Constitutive Model Parameters of X12 Ferritic Heat-Resistant Steel Using Reverse Optimization Method

Due to the rapid development of parameter reverse technology and computational simulation technology, through a comparison of test data of finite element simulation and the confirmed model, the parameters of constitutive model delineating the mechanical properties of materials can be obtained by combining them with the application of the inverse method to solve problems, which are able to accurately reflect real mechanical properties of materials in a computer simulation test. Thus, the credibility of the model greatly is improved so as to reduce the research costs of related issues. The parameters to be reversed in the model are A, $m_{1}$, $m_{2}$, $m_{3}$, $m_{4}$, $m_{5}$, $m_{7}$, $m_{8}$ and $m_{9}$ in this paper. For the problems of the inverse method of material parameters, the method of optimization design is used to continuously optimize the parameters of the constitutive model until a set of parameters is obtained, which is able to minimize the gap of the mechanical properties between the real material and the confirmed model. The adaptive simulated annealing algorithm (ASA) reflects the optimization process of the optimization problem via the annealing process of the metal; each feasible solution is related to the different states of the metal. In a closed system with thermal equilibrium, when the free energy reaches the minimum state, the system is in equilibrium according to the first law of thermodynamics. Meanwhile, the corresponding feasible solution is the optimal solution of the optimization problem. The ASA algorithm is well suited for solving highly nonlinear problems with short running analysis codes, when finding the global optimum is more important than a quick improvement in the design. Theoretically, the simulated annealing algorithm can always find the global optimal solution of a problem [31].

In this paper, the adaptive simulated annealing optimization method is adopted as the identification strategy for minimizing the objective function. The parameter corresponding to the minimum error of the objective function is the optimal solution. The objective function is Equation (10) and the flow diagram is represented in Figure 9. Figure 10 and Figure 11 show the optimization curves and the convergence of the objective function, using the adaptive simulated annealing algorithm to identify parameters. As can be seen from Figure 10, the fluctuation trend of the parameter value becomes smaller and smaller when the iteration steps increase. It can be observed from Figure 11 that the objective function value gradually decreases until it approaches 0.035, indicating a good convergence. As a result, the parameter values corresponding to the optimal solution are shown in Table 2.
(10)$$O(f)\;=\;\frac{\sum_{i}^{n}{\left(\sigma_{i}^{\exp}\;-\;\sigma_{i}^{cal}\right)}^{2}}{\sum_{i}^{n}{\left(\sigma_{i}^{\exp}\right)}^{2}}$$
where $\sigma_{i}^{\exp}$ is the flow stress obtained by the experiment and $\sigma_{i}^{cal}$ is the flow stress predicted by the Hansel–Spittel model.

Eventually, the elevated temperature deformation constitutive model of X12 ferritic heat-resistant steel is solved by the reverse optimization method:(11)$$\sigma \;=\;\begin{array}{c} 2.5347\;\times \;10^{5} \end{array}e^{-0.004253T}\varepsilon^{-0.0621}{\dot{\varepsilon }}^{-0.2955}e^{\frac{-0.01875}{\varepsilon }}{\left(1\;+\;\varepsilon \right)}^{0.0015T}e^{-1.3531\varepsilon }{\dot{\varepsilon }}^{0.00038T}T^{-0.4075}$$

### 3.3. Validation of Constitutive Model for X12 Ferritic Heat-Resistant Steel

The accuracy of the flow stress model established in this paper can be reflected in terms of the correlation coefficient [32] and the root mean square (RMS) [33]. The formulas are presented as follows:(12)$$R\;=\;\frac{\sum_{i=1}^{N}\left(\sigma_{e}^{i}\;-\;{\bar{\sigma }}_{p}\right)(\sigma_{p}^{i}\;-\;{\bar{\sigma }}_{p})}{\sqrt{\sum_{i=1}^{N}{\left(\sigma_{e}^{i}\;-\;{\bar{\sigma }}_{e}\right)}^{2}}\sqrt{\sum_{i=1}^{N}{\left(\sigma_{p}^{i}\;-\;{\bar{\sigma }}_{p}\right)}^{2}}}$$
(13)$$RMS\;=\;\sqrt{\frac{1}{N}\sum_{i=1}^{N}{\left(\sigma_{e}^{i}\;-\;\sigma_{p}^{i}\right)}^{2}}$$
where $\sigma_{e}$ represents the flow stress obtained by experiment, ${\bar{\sigma }}_{p}$ represents the average flow stress predicted by Hansel–Spittel model and N is total number of all data. The closer R is to 1, the smaller RMS is, which indicates that the model is more accurate in predicting experimental data. In order to verify the accuracy of the model, the independent data are added, which are not used to build the model under other conditions [34]. A set of data are strain rate of 1 s^−1^ and temperatures of 1125 °C, 1175 °C, 1225 °C, respectively. The calculated results are shown in Figure 12. The degree of the oblique line is 45 in the figures. The scattered points of the predicted flow stress value and the test value are distributed near the 45-degree line. The closer the distribution is to the 45-degree line, the better the prediction effect is. The farther away from the 45-degree line, the worse the predicted effect is. R as well as RMS of constitutive model constructed by the traditional calibration method are as follows: R = 0.8773, RMS = 7.338 × 10^5^ MPa. R, as well as RMS, of constitutive model constructed by the inverse optimization method and are as follows: R = 0.97833, RMS = 6.2707 × 10^5^ MPa. The results indicate that the discrete degree of the test points and prediction points in the constitutive model constructed by the traditional calibration method are higher than those obtained by the inverse optimization method. Figure 13 shows the stress and strain values predicted by the models, constructed through different methods, at a strain rate of 1 s^−1^ and temperatures of 1125 °C, 1175 °C and 1225 °C, respectively. It can be seen from the figure that the predicted stress–strain curve of the model solved by the reverse optimization algorithm is closer to the actual stress–strain curve than that identified by the traditional calibration method. Eventually, it can be proved that multi-inverse optimization method has a higher accuracy and better robustness in solving the model.

### 3.4. Comparison between Numerical and Experimental Results in Compression Test

For the sake of further verifying the robustness of the model, compression tests and numerical simulations were carried out on two samples with different shapes. One sample was a cylindrical sample with a height of 12 mm and a diameter of 8 mm. The other sample was a cylinder sample with a height of 12 mm and a diameter of 8 mm. It had a notch with a radius of 1 mm. A cylinder sample with a height of 12 mm and a diameter of 8 mm was compressed at 1100 °C/0.1 s^−1^. Another type of cylinder sample was compressed at 1225 °C/0.75 s^−1^. The Hansel–Spittle flow stress model of X12 ferritic heat-resistant steel was integrated into FORGE^@^ software to simulate the compression test of the material. Since there are no data for X12 heat-resistant steel in the material database, they needs to be created. The parameter values $m_{1}$, $m_{2}$, $m_{3}$, $m_{4}$, $m_{5}$, $m_{7}$, $m_{8}$, $m_{9}$ of the Hansel–Spittle model were input into the material database. Half of the cylindrical sample (Ø 8 × 12 mm) and the same cylindrical sample with a notch radius of 1 mm were modeled, as shown in Figure 14. The compressed deformation was 50%. The friction coefficient between dies and the billet was 0.05. The shear factor was 0.1 and the heat exchange coefficient was set to 10^4^ Wm^−2^K^−1^. The storage type was “Height” and the step length was 1 mm. Figure 15a,b shows the equivalent strain distribution of the workpiece after compression of the two models. It can be observed from the figure that the equivalent strain is not uniform, and the large strain is mainly distributed in the center of the workpiece, where dynamic recrystallization is most likely to occur. The places are mainly distributed in difficult-to-deform zones where the strain is small, such as parts were contact dies. The equivalent strain of the model using the reverse optimization method is higher than the value using the traditional calibration method. Figure 15c shows the displacement–load curves of experiments and simulation, respectively. It shows that displacement–load value simulated by the model using the reverse optimization method is generally lower than the experimental displacement–load value, but higher than the value obtained by the traditional calibration method. It can be seen from Figure 16a,b that the strain of the notch portion is higher than that of the surrounding cylindrical surface. It can be seen from Figure 16c that the displacement–load value simulated by the model using the reverse optimization method is close to the experimental displacement–load value, but greater than the value obtained by the traditional calibration method. The error was small compared with the actual situation, which verified the accuracy of the X12 ferritic heat-resistant steel flow stress model constructed by the inverse optimization method.

### 3.5. Hot Processing Maps of X12 Ferritic Heat-Resistant Steel

Gegel [35] regarded the thermal deformation process as a closed thermodynamic system and the total power dissipation was regarded as P, which was divided into two parts consisting of dissipation quantity G and dissipation co-quantity J. Its mathematical expression is as follows:(14)$$P\;=\;\sigma \dot{\varepsilon }\;=\;G\;+\;J\;=\;\int_{0}^{\dot{\varepsilon }}\sigma d\dot{\varepsilon }\;+\;\int_{0}^{\sigma }\dot{\varepsilon }d\sigma$$
where J is power dissipation by the evolution of the material’s microstructure during the hot working process, G is the total power consumed when the material is deformed. m is the ratio of J and G, which is called the sensitivity coefficient of strain rate. The Formula (15) is as follows:(15)$$m\;=\;\frac{\partial J}{\partial G}\;=\;\frac{\dot{\varepsilon }\partial \sigma }{\sigma \partial \dot{\varepsilon }}\;=\;\frac{\partial \ln\sigma }{\partial \ln\dot{\varepsilon }}$$

When the sensitivity coefficient of strain rate m is equal to 1, J reaches the maximum value. Therefore, the following formula can be derived:(16)$$\eta \;=\;\frac{J}{J_{\max}}\;=\;\frac{J}{\sigma \dot{\varepsilon }/2}\;=\;\frac{2m}{m\;+\;1}$$

The criterion of continuous instability of the dynamic material model was proposed by Ziegler [36]:(17)$$\xi (\dot{\varepsilon })\;=\;\frac{\partial \ln\left(\frac{m}{m\;+\;1}\right)}{\partial \ln\dot{\varepsilon }}\;+\;m\;<\;0$$
where $\xi (\dot{\varepsilon })$ represents flow instability coefficient. The distribution map is made using the range of strain rates and temperatures, and is called the rheological instability diagram. The rheological instability will occur in the negative value region while the positive region is the rheological stability region. The typical microscopic phenomena of rheological instability include mechanical twins, adiabatic shear bands, local deformation, torsion and other phenomena [37]. A hot processing map is formed by superimposing the instability diagram on the dissipation diagram. A hot processing map is a tool to evaluate the formability of materials. Thus, the “safe zone” and “flow instability zone” of processing map can be obtained [38,39,40].

Figure 17 presents hot processing maps of X12 ferritic heat-resistant steel under various strains. The green is the rheological instability zone and the values of the contour line are coefficients of power dissipation. It can be observed from Figure 17a that, as strain is equal to 0.1 and the temperature is 1200–1250 °C, the dissipation coefficient reaches the peak value of 27% in the range of 0.05–0.06 s^−1^, which is less than 30%. In combination with the flow stress curves in Figure 2, it can be seen that the probability of dynamic recrystallization is relatively small. The rheological instability zone is concentrated in the low strain rate and low temperature zone (temperatures of 1050–1125 °C and the strain rates of 0.05–0.81 s^−1^); elevated temperature and low strain rate zone (strain rates of 0.05 to 0.65 s^−1^ as well as the temperatures of 1200 to 1250 °C), as well as high strain rate and elevated temperature zone (temperatures range from 1120 to 1200 °C, as well as the strain rate from 0.65 s^−1^ to 5 s^−^^1^). As can be seen in Figure 17b, the power dissipation coefficient reaches its peak value in the elevated temperature and low strain rate zone with a maximum value of 45%, which indicates that this zone of the material is prone to experience the dynamic recrystallization mechanism. It is in the unstable region of the hot processing map; therefore, the hot processing of the workpiece under a strain of 0.3 should be avoided as much as possible. As can be seen in Figure 17c, at strain rates of 0.05–0.35 s^−1^ and temperature ranges of 1125–1220 °C, the coefficients of power dissipation are all greater than 30%. This zone is prone to experience dynamic recrystallization and is in a stable processing region, while other zones are unstable. According to the analysis in Figure 17d, the distribution of the stable zone is roughly the same as that in Figure 17c, with the temperature ranging from 1150 °C to 1125 °C and strain rate ranging from 0.05 to 0.62 s^−1^. From the overall trend, the peak value of power dissipation coefficient of X12 ferritic heat-resistant steel increases with the gradual increase in strain. The range where the dissipation coefficient is more than 30% increases gradually. The probability of dynamic recrystallization occurring is higher. At the same time, it can be seen that the higher dissipation coefficient is mainly distributed in the lower strain rate range, while the rheological instability zone increases gradually with the raise of strain. When strain is 0.1–0.3, the rheological instability zone concentrates mainly in the low temperature (1050–1125 °C) and low strain rate zone (0.05–0.81 s^−1^), as well as the high temperature and low strain rate zone (1200–1250 °C, 0.05–0.65 s^−1^). When the strain is greater than 0.3, the rheological instability zone is mainly distributed in the high strain rate zone ($\dot{\varepsilon }\;>\;0.62$ s^−1^). Briefly, the optimal process parameters of the material can be obtained as temperatures of 1150–1200 °C and strain rates of 0.05–0.62 s^−1^.

### 3.6. Mechanism of Microstructure Evolution of Characteristic Region in Hot Processing Map of X12 Ferritic Heat-Resistant Steel

In order to further research the microstructure evolution of each region in the hot processing map, the various regions are discussed in detail. Figure 18 shows the distribution of various regions under a strain of 0.7. Region A is the instability region where the coefficient of power dissipation is lower than that of other regions. In order to avoid defects in the process of thermal processing, the processing region should be in the non-instability region. Region B, C and D are non-instable regions and their coefficients of power dissipation are relatively high. The power dissipation coefficients are greater than 30%. Whereas, there are different microstructures in the four regions. Figure 19 shows the microstructure under different regions of the hot processing map. The microstructure of Region A corresponds to the deformed structure in Figure 19a–c. Under these conditions, the adjacent grains are seriously deformed due to stress concentration under the action of shear stress, which will result in intergranular cracking and transgranular cracking. Thus, in these zones, the formability of X12 ferritic heat-resistant steel is poor and it is not suitable to process the workpiece. Although Regions B, C and D are all safe zones with high power dissipation coefficients, they do not represent optimal windows. The microstructure of Region B in Figure 19d shows that the grain distribution is uniform, revealing an equiaxed grain state as the average grain size is 23.48 μm, which indicates that complete dynamic recrystallization occurs under these conditions. The microstructure of Region C corresponds to Figure 19e. From the figure, the grain distribution is not uniform under this deformation condition and there are remnants of the original deformed grain with a minimum grain size of 2.28 μm and a maximum of 33.05 μm. The percentage of dynamic recrystallization is less than 50% because the thermal activation energy is low, leading to a slow nucleation of dynamic recrystallized grains at a low strain rate and low temperature. Figure 19f corresponds to the microstructure of Region D and shows that the grain size of the microstructure is coarse and that small grains are distributed around the large ones under this deformation condition. On the one hand, the grains grow rapidly due to the high deformation temperature. On the other hand, new recrystallized grains appear at the grain boundary. The grain structure is also not uniform. Therefore, safety region B is the optimal processing zone through the comparison of microstructure. On the basis of the above analysis of microstructure, Figure 20 is obtained, which is the distribution map of characteristic regions in the hot processing map for X12 ferritic heat-resistant steel. Region A corresponds to the instability area where cracks or microcracks occur. Region B corresponds to the safe zone with a uniform grain distribution. Region C corresponds to the area where incomplete dynamic recrystallization occurs and Region D corresponds to the area where sharp grain growth occurs. In conclusion, the optimal parameters in hot forging of X12 ferritic heat-resistant steel are strain rates of 0.05–0.62 s^−1^ and temperatures of 1150–1200 °C. In this range, the desired formability of X12 ferritic heat resistant steel can be required.

## 4. Conclusions

In this paper, the hot deformation behavior of X12 ferritic heat resistant steel at high temperature was researched. The hot compression test of X12 ferritic heat resistant steel was performed on a Gleeble-1500D test machine at strain rates of 0.05–5 s^−1^ and temperatures of 1050–1250 °C. On the basis of the test data, flow stress curves were drawn and the Hansel–Spittle model of the material was constructed. Moreover, a hot processing map of X12 ferritic heat resistant steel was drawn based on the data. The conclusions are as follows:

Based on reverse optimization technology, a new method for determining the parameters of the material hot forming constitutive model was put forward in this paper. The Hansel–Spittle model of X12 ferritic heat resistant steel, solved by the reverse optimization method, is as follows:$$\sigma \;=\;\begin{array}{c} 2.5347\;\times \;10^{5} \end{array}e^{-0.004253T}\varepsilon^{-0.0621}{\dot{\varepsilon }}^{-0.2955}e^{\frac{-0.01875}{\varepsilon }}{\left(1\;+\;\varepsilon \right)}^{0.0015T}e^{-1.3531\varepsilon }{\dot{\varepsilon }}^{0.00038T}T^{-0.4075}$$

The values of R and RMS for X12 ferritic heat resistant steel were 0.97833 and 6.2707 × 10^5^ MPa. Combined with finite element technology, the constitutive model of X12 ferritic heat resistant steel proposed in this paper was implemented into the commercial finite element software FORGE^@^. By comparing the samples’ displacement–load curves obtained by high temperature compression tests and numerical simulations, the accuracy of the high temperature constitutive model of X12 ferritic heat resistant steel was verified.

In order to further research the optimal forming range of ultra-supercritical rotors, the hot processing map of X12 ferritic heat resistant steel was constructed according to the dynamic material model and the optimal deformation range of X12 ferritic heat resistant steel was obtained, which were temperatures of 1150–1200 °C and strain rates of 0.05–0.62 s^−1^. Moreover, on the basis of microstructure analysis, a characteristic region distribution diagram in the hot processing map of X12 ferritic heat resistant steel was established. Therefore, the microstructure and properties of X12 ferritic heat resistant steel in the hot forging process can be more accurately controlled.

## Figures and Tables

**Figure 1 materials-14-01958-f001:**
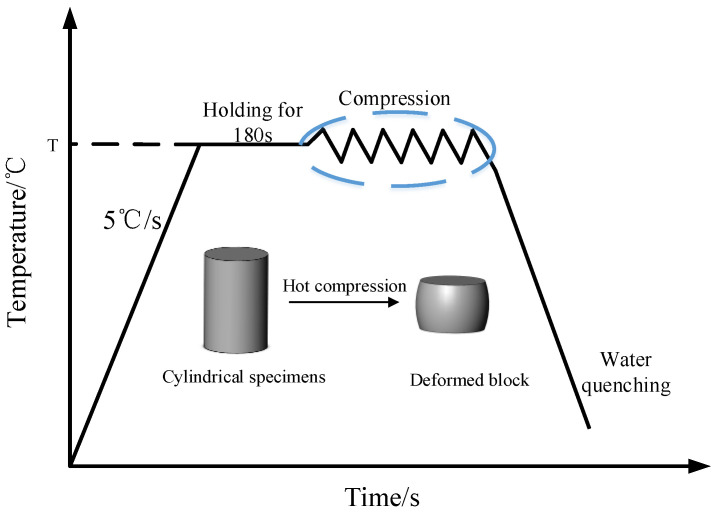
Deformation process of the hot compression experiments [23].

**Figure 2 materials-14-01958-f002:**
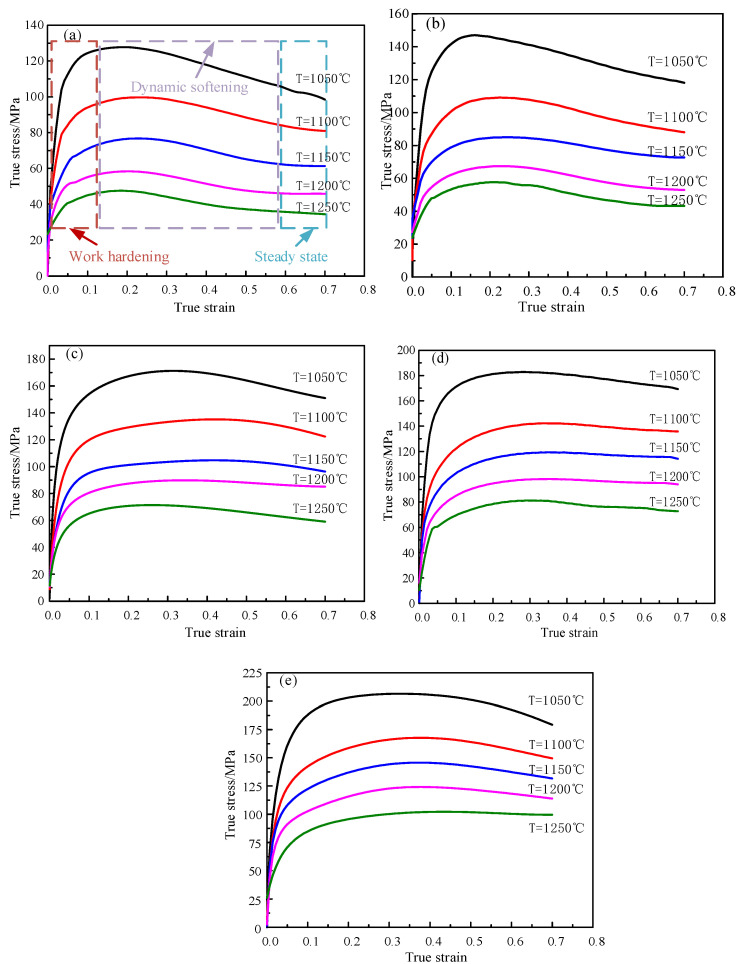
True flow stress curves of X12 ferritic heat-resistant steel under various deformation conditions: (**a**) $\dot{\varepsilon }$ = 0.05 s^−1^; (**b**) $\dot{\varepsilon }$ = 0.1 s^−1^; (**c**) $\dot{\varepsilon }$ = 0.5 s^−1^; (**d**) $\dot{\varepsilon }$ = 1 s^−1^; (**e**) $\dot{\varepsilon }$ = 5 s^−1^.

**Figure 3 materials-14-01958-f003:**
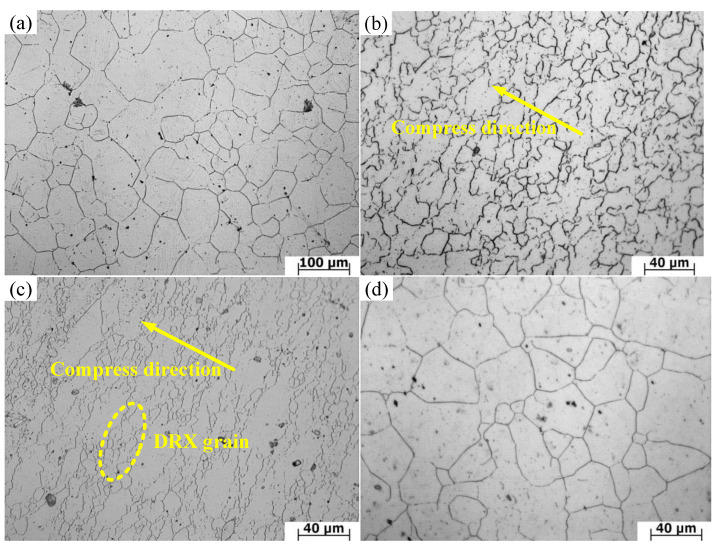
Microstructure of X12 ferritic heat-resistant steel under various deformation conditions: (**a**) original microstructure; (**b**) 1100 °C/0.05 s^−1^; (**c**) 1100 °C/0.5 s^−1^; (**d**) 1250 °C/0.5 s^−1^.

**Figure 4 materials-14-01958-f004:**
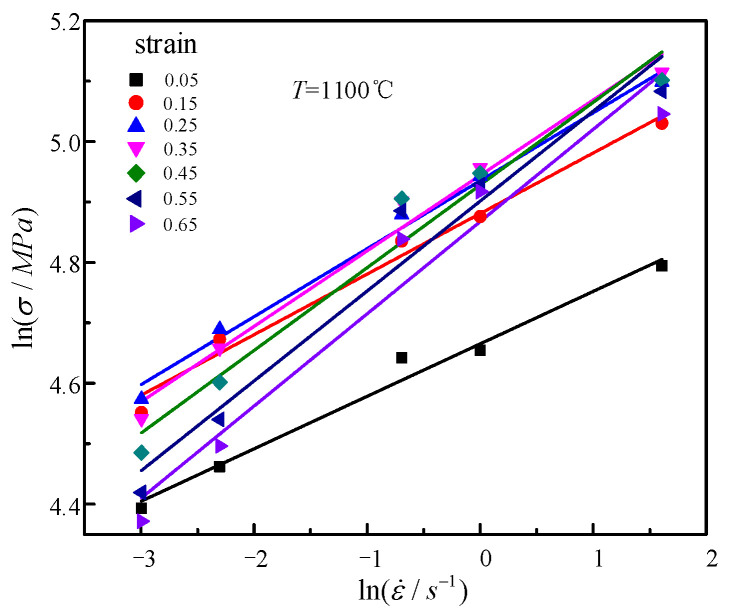
Relationships between $\ln\dot{\varepsilon }$ and lnσ at 1100 °C.

**Figure 5 materials-14-01958-f005:**
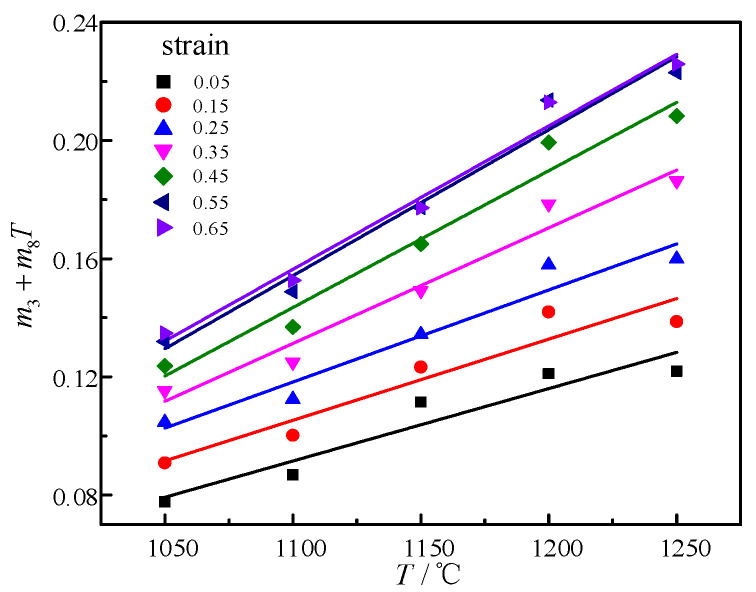
Relationships between $m_{3}\;+\;m_{8}T$ and T under different strains.

**Figure 6 materials-14-01958-f006:**
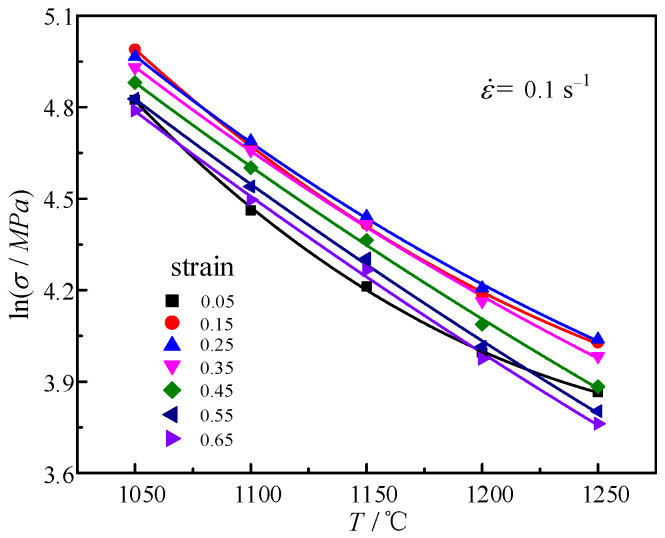
Relationships between lnσ and T at 0.1 s^−1^.

**Figure 7 materials-14-01958-f007:**
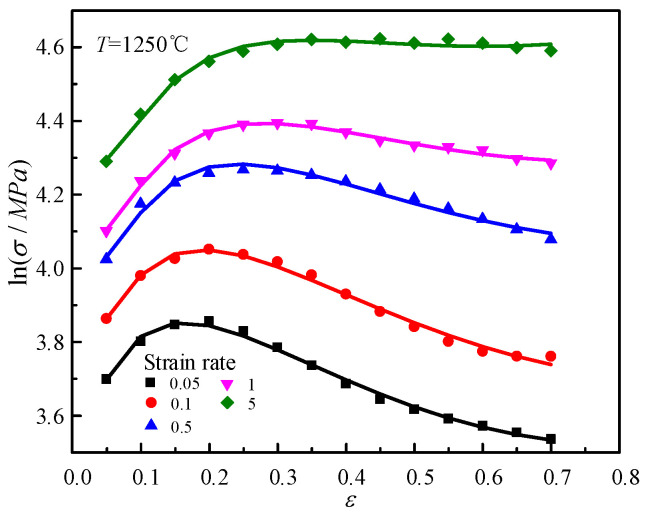
Relationships between lnσ and ε at 1250 °C.

**Figure 8 materials-14-01958-f008:**
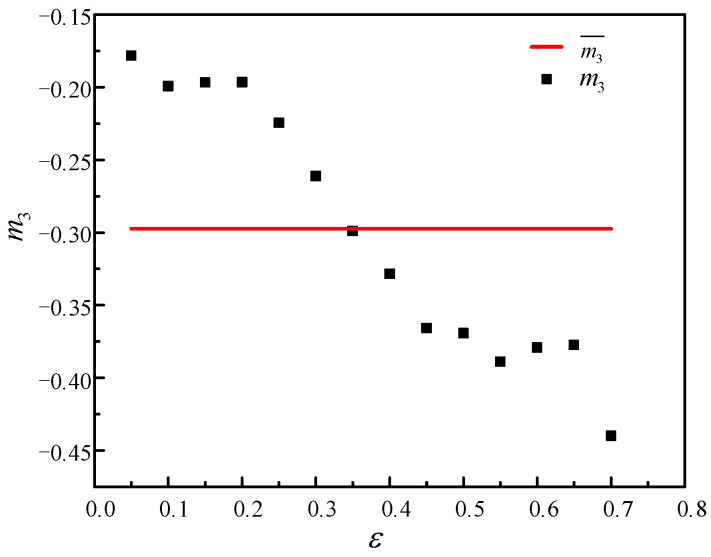
Relationships between $m_{3}$ and ε.

**Figure 9 materials-14-01958-f009:**
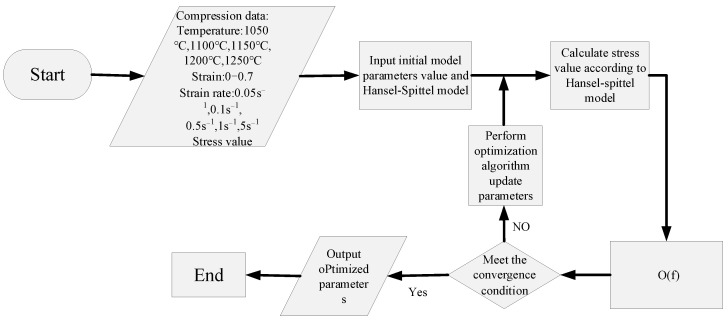
Flow chart of solving optimal parameters of material constitutive model by using adaptive simulated annealing algorithm.

**Figure 10 materials-14-01958-f010:**
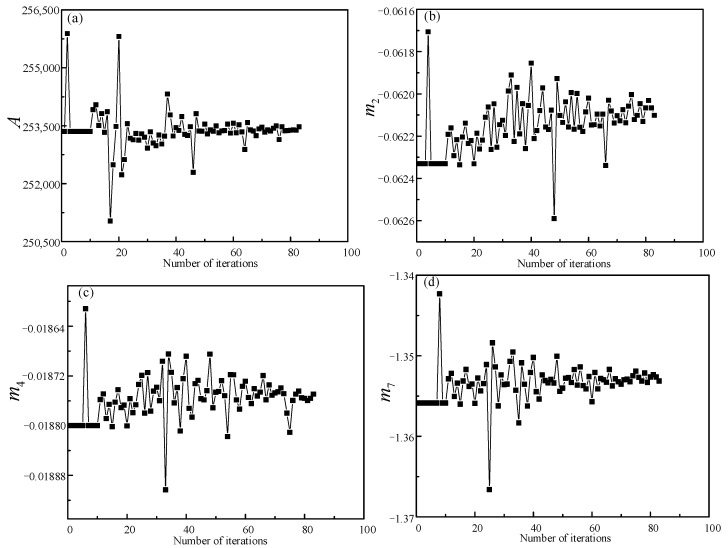
Optimization curves of parameters: (**a**) A; (**b**) $m_{2}$; (**c**) $m_{4}$; (**d**) $m_{7}$.

**Figure 11 materials-14-01958-f011:**
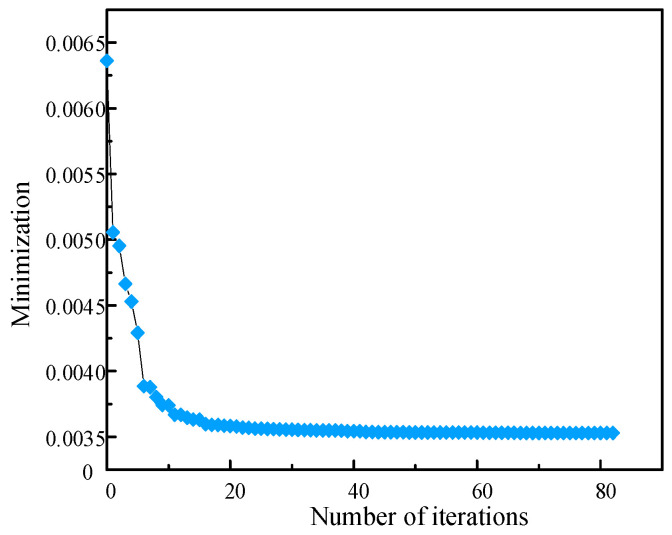
Optimization curve of objective function.

**Figure 12 materials-14-01958-f012:**
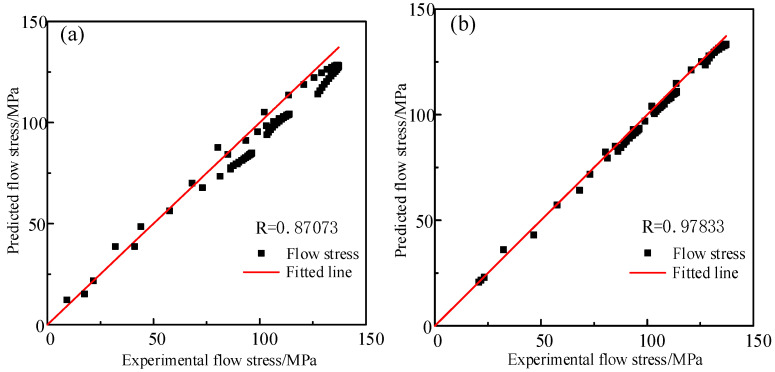
The relationship between experimental flow stress values and predicted flow stress values under different solution methods: (**a**) traditional calibration method, (**b**) adaptive simulated annealing optimization method.

**Figure 13 materials-14-01958-f013:**
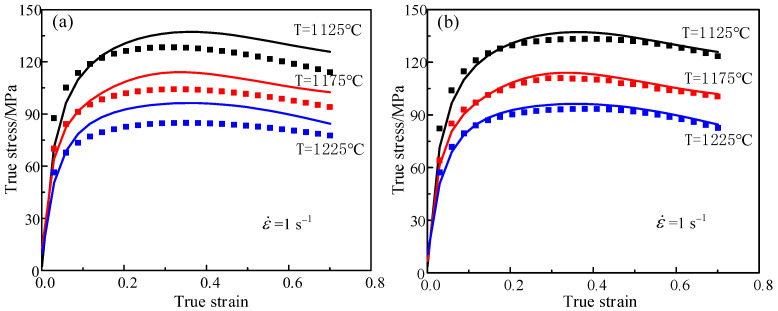
Comparative plots for the experimental results (lines) and fitted values of constructed model under different solution methods: (**a**) traditional calibration method, (**b**) adaptive simulated annealing optimization method.

**Figure 14 materials-14-01958-f014:**
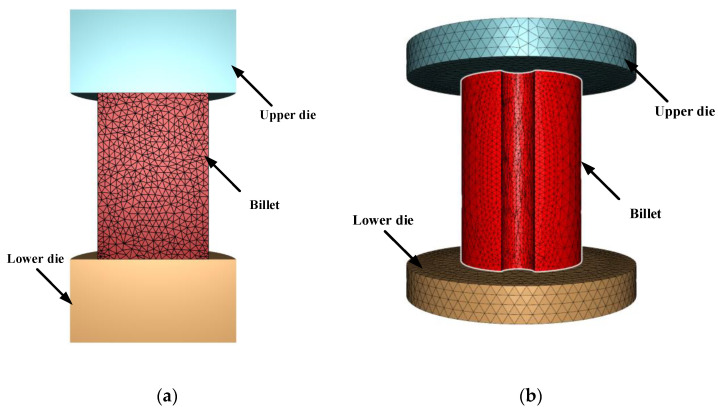
Models of different shapes: (**a**) Ø 8 × 12 mm half sample model; (**b**) the cylindrical sample (Ø 8 × 12 mm) with a notch radius of 1 mm.

**Figure 15 materials-14-01958-f015:**
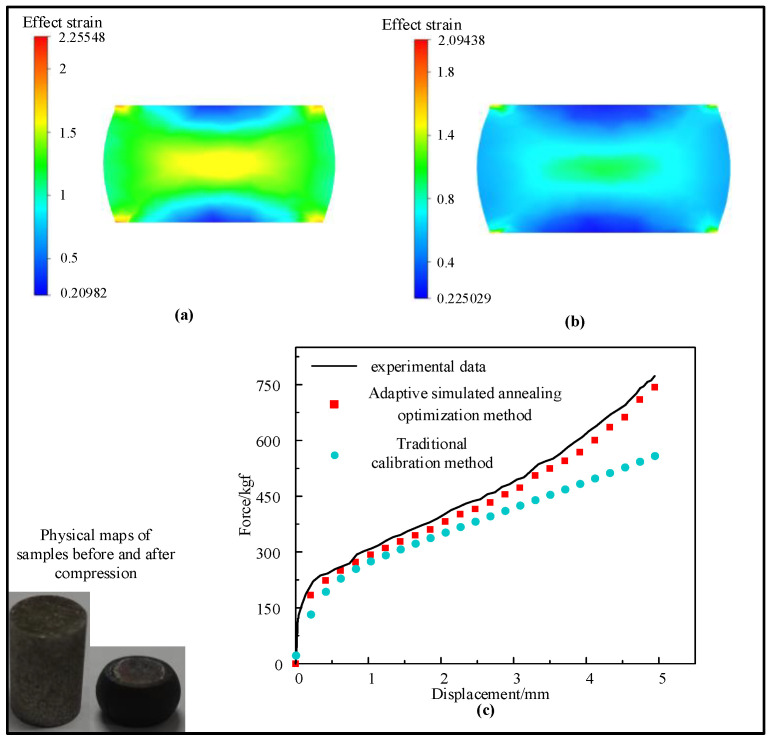
Simulation of cylindrical sample (Ø 8 × 12 mm): (**a**) the distribution of strain at 1100 °C/0.1 s^−1^ with adaptive simulated annealing optimization method; (**b**) the distribution of strain at 1100 °C/0.1 s^−1^ with the traditional calibration method; (**c**) force vs. displacement plot of experimental and predicted values (Hansel–Spittel model of two methods) at 1100 °C/0.1 s^−1^.

**Figure 16 materials-14-01958-f016:**
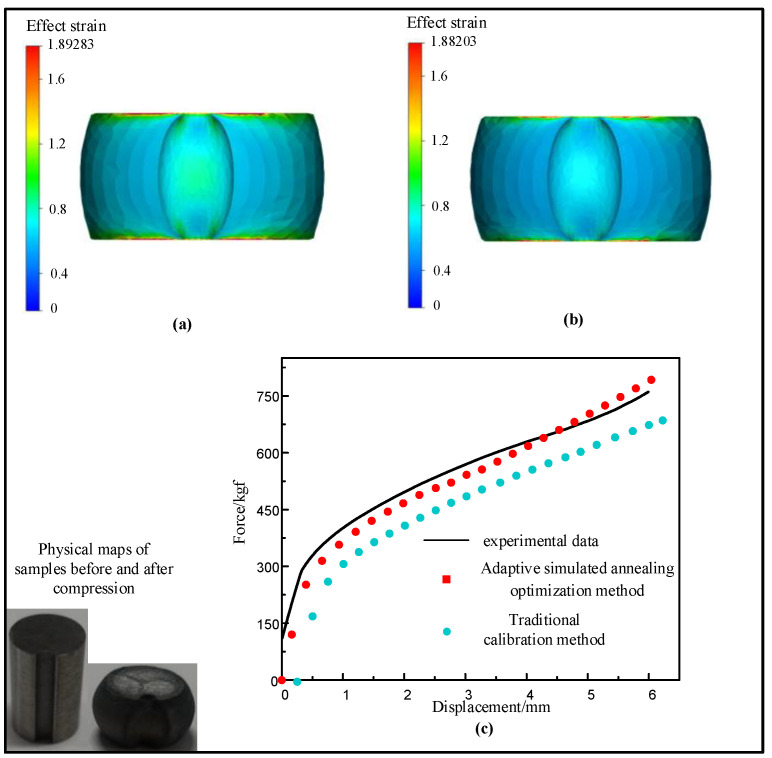
Simulation of cylindrical sample (Ø 8 × 12 mm) with a notch of 1 mm radius: (**a**) the distribution of strain at 1225 °C/0.75 s^−1^ with adaptive simulated annealing optimization method; (**b**) the distribution of strain at 1225 °C/0.75 s^−1^ with the traditional calibration method; (**c**) force vs. displacement plot of experimental and predicted values (Hansel–Spittel model of two methods) at 1225 °C/0.75 s^−1^.

**Figure 17 materials-14-01958-f017:**
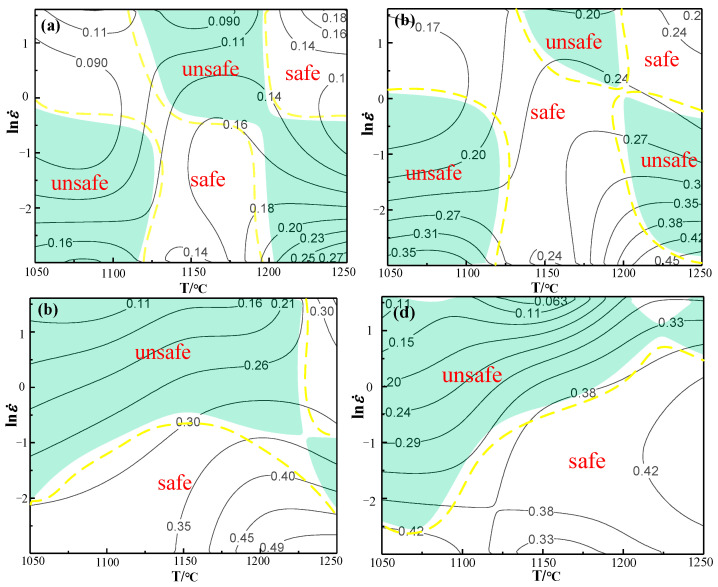
Hot processing maps of X12 ferritic heat-resistant steel at different strains (**a**) ε = 0.1; (**b**) ε = 0.3; (**c**) ε = 0.5; (**d**) ε = 0.7.

**Figure 18 materials-14-01958-f018:**
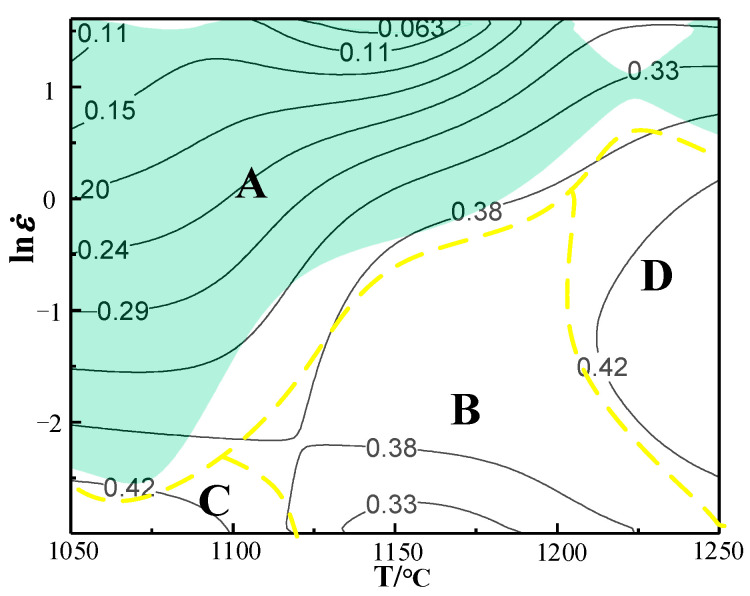
Different regions of the thermal processing map under the condition of a strain of 0.7.

**Figure 19 materials-14-01958-f019:**
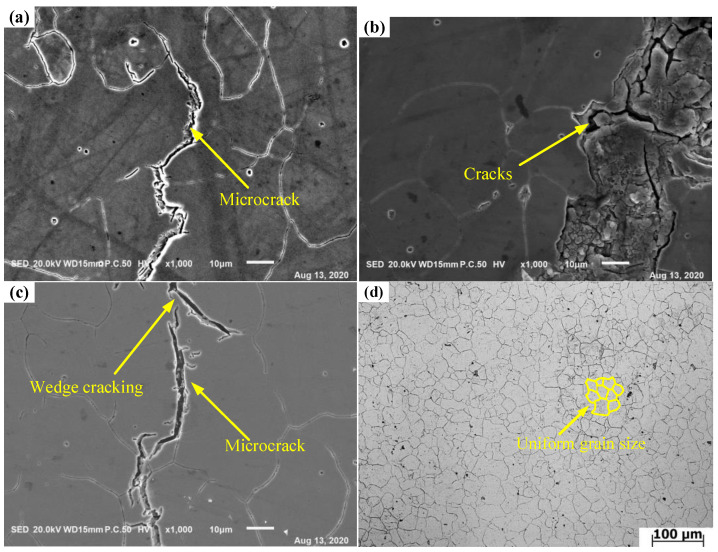
Microstructure of X12 alloy steel under various deformation conditions. (**a**) 1100 °C/1 s^−1^ (Region A); (**b**) 1150 °C/1 s^−1^ (Region A); (**c**) 1200 °C/5 s^−1^ (Region A); (**d**) 1150 °C/0.5 s^−1^ (Region B); (**e**) 1100 °C/0.05 s^−1^ (Region C); (**f**) 1250 °C/0.5 s^−1^ (Region D).

**Figure 20 materials-14-01958-f020:**
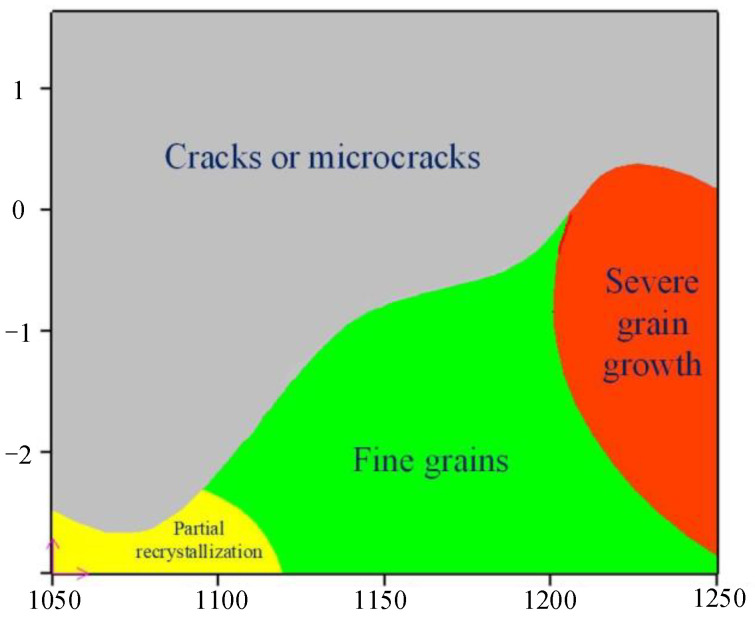
X12 heat-resistant steel characteristic area distribution map in hot processing map.

**Table 1 materials-14-01958-t001:** Chemical composition of X12 ferritic heat-resistant steel (mass percentage: wt.%).

C	Cr	Mo	Nb	V	W	Ni	Mn	N
0.188	11	1.029	0.069	0.207	0.95	0.744	0.42	0.055

**Table 2 materials-14-01958-t002:** Hansel–Spittel model parameters identified by reverse optimization.

*A*	*m* _1_	*m* _2_	*m* _3_	*m* _4_	*m* _5_	*m* _7_	*m* _8_	*m* _9_
2.5347 × 10^5^	−4.253 × 10^−3^	−0.0621	−0.2955	−0.01875	0.0015	−1.3531	3.8 × 10^−4^	−0.4075

## Data Availability

The data presented in this study are available on request from the corresponding author. The data are not publicly available due to these data are part of ongoing research.

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
