# Peer review of "Identification of the Constitutive Model Parameters by Inverse Optimization Method and Characterization of Hot Deformation Behavior for Ultra-Supercritical Rotor Steel"

_materials, 2021, doi:10.3390/ma14081958_

Round 1

Reviewer 1 Report

In this paper, the hot deformation behavior for ultra-supercritical rotor steel alloy, X12, at elevated temperature were analyzed through thermal compression tests. The compression tests were performed at a strain rates of 0.05-5 s-1 and deformation temperatures of 1050-1250°C. The flow behavior of this alloy was modeled via a constitutive model. Combined with finite element technology, the constitutive model was embedded into FORGE@ software to predict the elevated temperature deformation behavior of the investigated alloy. Finally, the optimal processing parameters range of the investigated alloy was proposed. The present paper is interesting, however, to be accepted for publication the following comments need to be addressed.

  • The proficiency of the language needs a more improvement in the manuscript
  • The introduction section needs to be improved. It should summarize the existing research related to your work. In this study it is long. In addition, it is recommended to cite new and related articles of the current journal.
  • The constructed model was fed into the simulation software, then the simulated results was compared with the experimental one at 1100°C /0.1 s-1. The expected results should be great because the authors used the experimental data of 1100°C /0.1 s-1 to construct the model that was fed into the software. The authors didn’t verify the constructed model or the simulation at new conditions, for example 1175°C /0.35 s-1. The constructed model should be tested by other conditions unmodeled conditions to prove its predictability. So, you need to perform a modelled experiment then compered the results with the modelled one or you can use the cross-validation technique as in the following paper.
  1. A.O. Mosleh, P. Mestre-Rinn, A.M. Khalil, A.D. Kotov, A. V. Mikhaylovskaya, Modelling approach for predicting the superplastic deformation behaviour of titanium alloys with strain hardening/softening characterizations, Mater. Res. Express. 7 (2020). doi:10.1088/2053-1591/ab59b5.
  • The Conclusion should be rewritten. It should be more concise.

Reviewer 2 Report

The authors use a numerical inverse optimisation method to fit a Hans Spittel constitutive model to compressive experiments. In my optinion, the manuscript has several faults, which prohibit the pubilication in the pesent form:

major issues:
- There is extensive language editing necessary.  The mansucript is hard to follow in the present form.

- In my opinion, the inverse optimisation of the model should be validated properly. Optimally, there is independent experimental data with a different geometry to validate the parameterised model.

The formatting of the manuscript needs to be improved. There are several formatting problems, which include (amongst others):

- The equations on page 3 need to be formatted properly.

- Figure 3: Why is the scale of the microsections not constant from a -d?

- Figure 4-6: The reader cannot differentiate between the individual plots   - Figure 18: I don't think that a whole page is necessary to explain a simple iterative optimsation approach.

- Figure 18: The labels on the plot are missing

These issues should be improved in a major revision of the article.

Round 2

Reviewer 1 Report

Thank you for taking my comments in your consideration, the present manuscript should be accepted in present form.

Reviewer 2 Report

The authors have implemented my suggestions and improved the manuscript considerably.